# Awareness, Attitudes and Willingness to Donate Biological Samples to a Biobank: A Survey of a Representative Sample of Polish Citizens

**DOI:** 10.3390/healthcare11202714

**Published:** 2023-10-11

**Authors:** Łukasz Pronicki, Marcin Czech, Mariusz Gujski, Natalia D. Boguszewska

**Affiliations:** 1Department of Public Health, Medical University of Warsaw, 02-091 Warszawa, Poland; 2Institute of Mother and Child, 01-211 Warszawa, Poland

**Keywords:** awareness, attitudes, biobanking, biobank, tissue donation, willingness

## Abstract

Biotechnology is developing at an ever-increasing pace, and the progressive computerization of health care and research is making it increasingly easy to share data. One of the fastest growing areas is biobanking. However, even with the best equipment and the best trained staff, a biobank will be useless without donors. For this reason, we have decided to gauge Polish citizens’ awareness and attitudes towards biobanking and their willingness to donate biological samples. For this purpose the survey was conducted among a nationwide group of 1052 Poles aged 18 and over where the totals for gender, age and place of residence were selected according to their representation in the total population of adult Poles. The survey was conducted using the Computer Assisted Web Interview (CAWI) technique. Approximately two thirds of respondents *N* = 701 (66.6%) indicated that they had heard of scientific studies in which samples of biological material such as blood, saliva or urine are collected. More than half of respondents (*N* = 613, 58.3%) had a positive opinion regarding scientific research in which samples of biological material are taken. Only *N* = 220 (20.9%) of respondents had previously encountered the term biobanking. More than a half *N* = 687 (65.3%) of respondents would participate in a scientific study that biobanked biological material and health information. Almost half of the respondents (48.0%) would like specific consent to be used in biobanking. In our study we observed a negligible correlation between socio-demographic factors and a willingness to donate biological material to a biobank. Considering the results presented above, the level of knowledge and awareness of biobanks, and their role in scientific research and the health care system, among Polish citizens is low and requires education and information activities.

## 1. Introduction

### 1.1. Role of International Organizations in Biobanking

Biobanking is one of many key processes related to the development of the health care system [1]. Collecting and storing biological samples allows for the building of repositories that can be used to search for new diagnostic and therapeutic methods, thus contributing to the development of health care systems both in Poland, Europe and the world. Integrated approach to research allows to uncover intricate connections between genetic factors, environmental influences, and the development of various diseases, paving the way for more personalized and effective healthcare strategies [2]. Biobanks are high-tech scientific units that collect human biological material and a lot of data related to the health of the donors of this material. These facilities are essential for the development of personalized medicine, that is, a branch of medicine that takes into account the individual needs of the patient, taking into account his or her characteristics at the level of the genome, metabolome and environment in which he or she lives. The development of biobanking and personalized medicine should, by design, seek to change treatment algorithms, where broad preventive health care comes to the fore [3,4]. The international organization for biobanking stakeholders is the International Society for Biological and Environmental Repositories (ISBER). ISBER’s mission is to increase expertise and quality among biobanks and biorepositories around the world [5]. As part of its activities, it provides education, develops best practices and works to promote biorepositories to the public [6,7]. It also organizes symposia and conferences where professionals and scientists can expand their knowledge and discuss current issues [8]. ISBER also engages in issues related to the involvement of potential donors by publishing publicly available guidelines on the ethical, legal and social aspects of collecting biological material [8,9]. As of 2023, ISBER has 1059 members from 60 countries, including 412 individual members, 137 small organizations, 63 medium-sized organizations, 41 large organizations and 1 very large organization [8]. In Europe, the organization for biobanking stakeholders is the Biobanking and Biomolecular Resources Research Infrastructure—European Research Infrastructure Consortium (BBMRI—ERIC), which was established in 2013 [10]. The enabling document for the establishment of BBMR-ERIC was Council Regulation (EC) No 723/2009 of 25 June 2009 on the Community legal framework for a European Research Infrastructure Consortium (ERIC), introduced by the European Union [11]. This document allowed the establishment of consortia among EU member states with the main task of developing research infrastructure to promote innovation, knowledge and technology transfer. It is worth mentioning that the main task of an ERIC established under this regulation should be the establishment and operation of research infrastructure on a non-profit basis. In order to promote innovation and the transfer of knowledge and technology, the ERIC should be allowed to carry out limited economic activities, provided that they are closely related to its main task [11]. BBMRI-ERIC is a consortium currently comprising 23 countries and 1 international organization [10,12]. The goal of BBMRI-ERIC is to bring together all the main stakeholders from the biobanking field—researchers, biobankers, industry and patients—to develop biomedical research [10,12]. The consortium offers quality management services, support with ethical, legal and societal issues, and a number of online tools and software solutions for biobankers and researchers. The goal of BBMRI—ERIC is to make new treatments possible [10,12]. BBMRI is also involved in a number of initiatives related to the development of biobanking, some of the most important include European Biotech week, part of the international Global Biotech week initiative, BioTechX [13] and the preparation and development of a number of guidelines for biobanks. BBMRI—ERIC is also very involved in work on the ethical, legal and social aspects of biobanking; as part of its activities it conducts webinars [13] and publishes guidelines on such issues as obtaining and biobanking material from underage donors, or drawing attention to a number of aspects related to the protection of donors’ personal data [14].

### 1.2. Status of Biobanking in Poland

In Poland, the human biological material is stored for research purposes at almost every medical university in the country, and in most scientific institutions carrying out research projects in the field of biomedicine [15]. In 2016, Poland joined BBMRI-ERIC to support the development of biobanking in Poland. Currently, as part of its BBMRI-ERIC membership, Poland is carrying out a number of tasks, including standardizing all sample procedures as well as data collection and storage; developing a universal IT infrastructure; promoting common quality control standards for all biobanks in the national network; analyzing the ethical, legal and societal needs related to research on human biological material; informing about the idea of biobanking human samples for research purposes among the public; and building and developing, since 2017, the Polish Network of Biobanks (PNB) [15,16,17]. The PNB’s mission is to integrate the entities involved in biobanking biological material, mainly from humans, and to facilitate the establishment of scientific collaborations. PNB members and observers can benefit from substantive support in developing their biobanks and biorepositories, as well as exchange information about their biological samples on a central IT platform [18]. PNB currently includes more than 50 units spread across the country, 6 of which are certified by BBMRI [19]. A survey of biobanks/biorepositories in Poland was conducted in 2021. In total, 65 of the 400 to whom the questionnaire was sent out fully completed the survey. A total of 60 of them collect human biological material and/or related data. Units collecting human biological material collect blood samples, cells, tissue samples and nucleic acids. Almost two thirds (39) of these units collect patient data related to the biological material being collected. It is also worth noting that 7% of units have collections larger than 100,000 samples, 17% of units have collections in the range of 10,001 to 100,000 samples and 28% of units have collections in the range of 1001 to 10,000 samples. The remainder have smaller collections [20]. In 2023, based on information available on the PNB website, it is possible to search for more than 500,000 samples from more than 140,000 donors in the eight biobanks that contribute their data to the system. It is possible to find 27 collections of biological material in the search engine [21]. In Poland, around 90% of biobanks belong to the public sector, and 60% of biobanks have declared their willingness to cooperate and share their biological material and data [20].

### 1.3. Ethical, Legal, and Social Issues of Biobanking

While biobanks are an significant resource for scientific research and ultimately the healthcare system, there are also ethical, legal and social concerns about their use, such as donor privacy and confidentiality, data protection and the commercialization of genetic products or technologies [22,23,24,25,26]. Conducting scientific research using biobanked biological material is incredibly challenging from an ethical perspective. Developments in biotechnology and medicine theoretically allow the use of biological material stored in biobanks in new scientific projects. Such action, however, contradicts the ethical principles we have developed as a society. It is unacceptable to use material received from a donor without the donor’s prior consent [23,27], and for this reason there have been moves in the scientific community to develop and expand the new types of consent that the donor of the material gives [23,24]. However, this still raises a number of questions, because a potential donor, despite giving broad consent, would not necessarily want to agree that his material be used in a particular study. On the other hand, not using already collected material also raises ethical concerns; not using it and collecting material from new donors exposes them to participation in unnecessary treatments or procedures. For this reason, modern methods of consent collection are constantly being developed and evaluated [23,24]. Legal issues also pose significant challenges for biobanking and conducting scientific research using collected biological material. The development of new technologies is often much faster than the development of legislation [22,24]. For this reason, we often face challenges in determining how a particular scientific study should be treated in legal terms and which pieces of legislation to apply [22]. This poses a major challenge for researchers and ethics committees. In Polish law, biobanking itself is also unregulated, which significantly hinders the development of biobanking. On the one hand, there are no legal regulations that say specifically how things should be done, and on the other hand, biobanks should be able to rely on regulations and work according to existing rules [22]. Another prominent issue is concerning the social aspects of biobanking. Public awareness and acceptance of research and biobanking is crucial for increasing donor commitment to donating their biological material [26,28]. For potential donors transparency, securing their data, confidentiality and the expediency of potential projects are important [26,29]. Society has much more confidence in the public than in the private sector [25]. Nevertheless, we are currently facing a big challenge. The development of modern technologies and medicinal products will probably always require the participation of the commercial sector at some stage [25,30]. Therefore, if we want the world’s repositories to be used in the search for new medicinal products, we must clearly define the rules of cooperation between public biobanks and the commercial sector. The rules of this cooperation raise many concerns among the public, for this reason it is necessary to be in dialogue with and develop solutions to safeguard the interests of potential donors [30,31]. For biobanks to be useful, they must be fed with information from donors who are willing to participate in scientific research and donate their biological material. Therefore, it was decided that we should conduct a survey on a representative group of adult Poles to determine their awareness, attitudes and willingness to donate their biological material and medical data to a biobank. We also decided to check what the opinion of Polish citizens is on informed consent and the level of their trust in biobanking stakeholders. The information obtained will allow for a better understanding of the needs, expectations and concerns of potential donors and will help build conditions for effective communication between biobank staff and potential donors.

## 2. Materials and Methods

### 2.1. Study Design

Our aim was to determine the public awareness of biobanking in Poland, to determine the willingness of potential donors to donate their biological material and to determine what factors influence their decisions. We also decided to check the level of public trust in particular stakeholders related to biobanking. For this reason, we decided to conduct a survey on a representative group of Poles. In order to collect the data, we decided to use the Computer Assisted Web Interviews (CAWI) method because of its effectiveness and its ability to reach the survey population. A questionnaire was developed and it was then commissioned to Ariadna, a national independent research panel. Received data were statistically analyzed.

### 2.2. Participants and Setting

The survey was conducted between January and February 2023, among a nationwide group of 1052 Poles aged 18 and over where the totals for gender, age and place of residence were selected according to their representation in the total population of adult Poles. Each respondent was subject to verification upon registration and was guaranteed full anonymity and confidentiality of their personal information.

The survey was not conducted using random selection methods, where respondents are collected from ad hoc roundups via pop-ups displayed on the Internet to random people, or via mass mailings or online surveys.

### 2.3. Research Tools

The questionnaire contained 23 questions about Poles’ knowledge, opinions, beliefs and attitudes toward biobanking and scientific research in which biological material is collected. The questionnaire for the survey was based on open-access questions used in the 2010 Eurobarometer survey [32] and questions used in the survey on public willingness to participate in biobanking in Switzerland published in 2021 by C. Brall et al. [33]. The questions were translated into Polish, consulted by experts and adapted to the specifics of Polish society. The questionnaire examined the level of awareness of biobanking, factors influencing the decision to donate biological material to a biobank, determined what the public’s expectations are regarding the supervision of biological material and data in biobanks, and examined attitudes and determined what factors influence parents’ decisions to donate their children’s biological material to a biobank. This publication only describes the areas of the level of awareness, the factors affecting the decision to donate material, and identifies public expectations.

### 2.4. Data Collection

The survey was conducted using the CAWI method with the help of Ariadna, a nationwide survey panel company. The Ariadna company allows researchers to carry out nationwide studies, surveys, and experimental research embracing the highest standards of rigor and integrity. During the implementation of the survey the Ariadna panel had a current and valid Interviewer Quality Control Program certificate [34] confirming the high quality of the research services provided, which is issued on the basis of an independent audit carried out annually by the Polish Association of Public Opinion and Marketing Research Firms.

Respondents receive an invitation to complete the survey sent via email, to the email address they provided when registering with the Ariadna panel. Respondents receive a message with a coded and personalized link to the survey. Real people with established identities took part in the survey. Upon completion of the survey, Ariadna provided a cleaned and anonymized dataset.

### 2.5. Ethical Issues

On 16 January 2023, through decision No. AKBE/3/2023, the study and the questionnaire received a positive opinion from the Ethics Committee at the Medical University of Warsaw. The survey data was completely anonymous and did not allow any identification of individual respondents.

### 2.6. Data Analysis

Continuous variables were summarized using mean and standard deviation. In addition, median, interquartile range, range and kurtosis were presented. For nominal variables, counts and percentages were used to summarize. Spearman’s Q was used for correlation analysis. A *p*-value (Hollander and Wolfe) was used to assess the significance of correlations. Due to multiple comparisons, a Benjamini–Hochberg correction was applied to the *p*-value. The relationship was considered significant at *p* < 0.05. The analysis was conducted in the R language environment.

## 3. Results

### 3.1. Demographic Characteristics of Respondents

One thousand and fifty-two respondents (*N* = 1052) took part in the survey. The average age of respondents was 46.37 years (SD = 15.92). Slightly more women (*N* = 567, 53.9%) participated in the survey than men (*N* = 485, 46.1%). More than half of the respondents are in a relationship (*N* = 573, 54.5%), and about a third of the participants said they were single (*N* = 310, 29.5%). The remaining respondents are divorced (*N* = 108, 10.3%) or widowed (*N* = 61, 5.8%). Most respondents have a university degree (master’s degree *N* = 357, 33.9% or bachelor’s degree *N* = 87, 8.3%). Most respondents live in rural areas (*N* = 390, 37.1%). Detailed demographic characteristics of the respondents are shown in Table 1.

### 3.2. Awareness of Research Using Biological Material and Biobanking

In 2023, *N* = 701 (66.6%) respondents indicated that they had heard of scientific studies in which samples of biological material such as blood, saliva or urine are collected. Only *N* = 146 (13.9%) respondents were unsure or could not remember if they had ever heard of such research. Only *N* = 220 (20.9%) of respondents had previously encountered the term biobanking, and as many as *N* = 674 (64.1%) had never heard of it before (Table 2).

More than half of respondents (*N* = 613, 58.2%) have a positive opinion regarding scientific research in which samples of biological material are taken (*N* = 441, 41.9%—rather positive, *N* = 172, 16.3%—positive), *N* = 371 (35.3%) respondents have no opinion on the subject and only *N* = 68 (6.5%) have a negative opinion (*N* = 50, 4.8%—rather negative, *N* = 18, 1.7%—negative) (Table 3).

The survey asked respondents to indicate which of the proposed definitions they thought best described the biobanking process. *N* = 441 (41.9%) respondents indicated that in their opinion, biobanking is a process in which body fluid or tissue samples, genetic data and medical data (medical history, laboratory results, etc.) are collected and stored in order to better understand health and disease progression. As many as *N* = 390 (37.1%) respondents did not know which of the proposed definitions best described the biobanking process (Table 4).

### 3.3. Knowledge and Opinion on Consent in Scientific Research

More than half (*N* = 619, 58.8%) of the respondents believed that they have to give consent for the biobanking their biological samples, but only *N* = 566 (53.8%) believed that they can withdraw such consent. It is noteworthy that as many as *N* = 360 (34.2%) respondents did not know whether they have to give consent for biobanking and *N* = 443 (42.1%) did not know whether they can withdraw the consent already given (Table 5).

The survey described to respondents a hypothetical situation in which a biobank would like to use their donated samples in one or more future scientific studies. They were then asked to indicate one of the following responses as to when and what they should give their consent to: each time the samples were to be used for a new project; depending on the scope/type of project; only once at the time the samples were donated; or no opinion. Most of the respondents believe that a biobank should ask for consent every time samples are to be used for a new project—specific consent (*N* = 505, 48.0%). According to *N* = 149 (14.2%) respondents, it is sufficient for them to give their consent once for the use of their samples in research projects—broad consent (Table 6).

Respondents considered the most important information needed to make a decision to donate their biological material are who will have access to their data and samples (*N* = 415 39.4%); what security measures are in place to ensure the privacy and protection of the samples (*N* = 407, 38.7%); and what research will be conducted using their samples (*N* = 386, 36.7%). Respondents considered the least important information to be who will receive financial benefits as a result of the research (*N* = 228, 21.7%) and who will receive non-financial benefits as a result of the research (*N* = 196, 18.6%) (Figure 1).

### 3.4. Willingness to Participate in the Study and Factors Influencing the Decision to Donate Biological Material

Nearly two thirds of respondents *N* = 687 (65.3%) would participate in a scientific study that biobanked biological material and health information (Table 7).

Those who agreed to participate in the study would be most willing to share samples that they can collect themselves (saliva, hair, urine, cheek swab) (*N* = 515, 75.0%), a blood sample (*N* = 470, 68.4%) or samples of biological material that must be collected by medical personnel (*N* = 384, 55.9%). Respondents are less willing to provide medical history (*N* = 318 46.3%), complete health forms (*N* = 301, 43.8%), family medical history (*N* = 174, 25.3%) or health data from sports apps (*N* = 170, 24.7%). Only *N* = 51 (7.4%) respondents considered sharing the information they post on their social media (Figure 2).

The greatest motivation for respondents to take part in a research study in which information about their health would be biobanked is the personal benefits they could gain from such a study (*N* = 430, 62.6%) or the benefits their family could gain (*N* = 374. 54.8%). The least motivating factor is a general sense of duty, with only *N* = 76 (11.1%) respondents taking this into account when making their decision (Figure 3).

Respondents who would participate in a scientific study conducted in a biobank would be most interested in receiving basic laboratory results such as blood counts (*N* = 801, 76.1%) and information on how their lifestyle affects their risk of contracting a disease (*N* = 725, 68.9%). They are least interested in general study results, with only slightly more than half (*N* = 581, 55.2%) wanting such information (Figure 4).

### 3.5. Concerns

Respondents indicated that the negative factor that would most likely influence their decision was the fear that their data would be used for commercial or marketing purposes rather than for research (*N* = 389, 37.0%). Respondents were also concerned about the confidentiality of their data (*N* = 350, 33.3%), and that the data they provided could be used to discriminate against them or their family members (*N* = 282, 26.8%). Factors such as a lack of time (*N* = 71, 6.7%), the effort involved in submitting biological material (*N* = 40, 3.8%) or general interest in scientific research (*N* = 39, 3.7%) influenced their decision to take part in a study the least (Figure 5).

### 3.6. Expectations for Research Using Biological Material and Biobanking

The majority (*N* = 591, 56.2%) of respondents would like their samples to be stored in a coded way—allowing them to be identified if needed. Most (*N* = 562; 53.4%) respondents would like to personally own the samples stored in the biobank, or would agree that the biobank itself should own the samples (*N* = 330; 31.4%). Respondents completely distrust the government in this area, with only *N* = 17 (1.6%) people indicating that the government should own the samples stored in the biobank (Table 8).

Respondents are most trusting of their general practitioner, with *N* = 213 (20.2%) people fully trusting and *N* = 390 (37.1%) trusting that their general practitioner would ensure the confidentiality and protection of the data and samples stored in the biobank if they had access to them. Another group are doctors in general and scientists working at universities and public institutes, trusted by about a third of respondents. The government, pharmaceutical companies and other profit-driven global and Polish companies were not trusted by respondents. It is noteworthy that more than half of the respondents (*N* = 581, 55.2%) completely distrust the government in the area of maintaining confidentiality and ensuring the security of data stored in the biobank (Table 9).

According to respondents, the responsibility for the storage of samples and related data should rest with the biobank’s board of directors (*N* = 567, 53.9%) or an independent committee of experts not associated with the biobank (*N* = 337, 32.0%). Respondents have the least confidence in a committee composed of representatives of the public (*N* = 155, 14.7%) or a mixed committee composed of experts and representatives of the public (*N* = 126, 12.0%) (Table 10).

### 3.7. Correlation Analysis

The study included a Sperman correlation analysis, which examined whether factors such as age, education, place of residence and gender are related to an awareness of scientific research in which biological material is used, knowledge of the term biobanking, opinion on scientific research and willingness to donate their samples to a biobank. The analysis showed a negligible correlation between the aforementioned factors (Table 11).

It was also checked whether there was a correlation between their opinion on scientific research, awareness about scientific research in which biological material is used, and knowledge of the term biobanking. An average correlation was shown between the opinion regarding scientific research and willingness to donate one’s biological material (Table 12).

## 4. Discussion

### 4.1. Public Awareness of Biobanks and the Scientific Research during Which Biological Material Is Collected

A similar distribution of the population was presented in a 2010 Eurobarometer survey on biotechnology [32], a survey conducted in Switzerland in 2020 [33] and a survey conducted in Latvia in 2019 [35]. According to the 2010 Eurobarometer survey on biotechnology, it was shown that, looking at the Central and Eastern European region, in Poland 28% of citizens had heard of biobanks, compared to 46% of Czech citizens, 34% of Slovak citizens, 34% of Lithuanian citizens and 31% of Hungarian citizens who had encountered the term [32]. The European average was 34%. In Germany, 30.8% of respondents in a 2018 survey had heard of biobanks [36]. In a 2019 survey in Latvia this percentage was 25.8% [6]. In Poland in 2023, around less than three quarters (71%) of medical students encountered the term biobank [37]. In 2013 in Poland, less than half (48%) of respondents associated a biobank with an institution related to medicine and 37% of respondents associated it with an institution related to scientific activities [38]. In the same survey, 39% of respondents had positive feelings about the word biobank, 35% mixed and 8% negative. Among 18% of respondents, the word biobank did not evoke any feelings [38]. According to the results of our survey, 20.9% of respondents in Poland in 2023 have heard of biobanking. However, in the question on indicating the definition of biobanking, as many as 41.9% of respondents indicated that, in their opinion, biobanking is a process in which body fluid or tissue samples, genetic data and medical data (medical history, laboratory results, etc.) are collected and stored in order to better understand the state of health and the course of disease. It should also be noted that in Poland, more than 66% of respondents indicate that they have heard of scientific research in which biological material is used, and 58.3% have a positive opinion of it. In Switzerland, 71% of the public has heard of such research and more than 60% have a positive opinion of it [33]. In Germany, 95.5% of respondents have a positive opinion about scientific research [36]. Comparing the results obtained from the present study with those obtained by other researchers, it can be concluded that both Polish society and other European societies have an awareness of scientific research in which biological material is collected, but are not fully aware that such material can be subject to biobanking and what this process consists of. We can also conclude that since the Eurobarometer 2010 survey in Polish society, there are no changes, either favorable or unfavorable, regarding the level of knowledge and awareness of biobanking.

### 4.2. The Approach to Informed Consent

Informed consent in the context of biobanking is a debated ethical and social issue [23,24,25,26]. Currently the use of specific consent is limited for many reasons—biological samples are used in many studies by many scientists working in different locations. Therefore, new models of consent are being sought and proposed [39,40]. Among medical school students in Saudi Arabia, 78% were aware that donating material to a biobank requires consent from the potential donor [41]. In the United States, in Colorado, 79% of respondents believed that the researcher must obtain consent from them to collect biological material for biobanking, 11% that there is no such requirement and 10% have no opinion [42]. In the 2010 Eurobarometer survey in Poland, 61% of respondents agreed with the use of specific consent while 22% agreed with the use of broad consent. The European average is 67% for broad consent and 18% for specific consent [32]. In Latvia, in 2019, 27.4% of respondents preferred broad consent and 62.2% of respondents preferred specific consent [35]. In 2013 in Poland, more than half of respondents on giving informed consent said it was very important that they would like to know what scientific projects their samples will be used for (59%) and how data and samples in the biobank are secured from illegal access (55%). In the same survey, a high percentage of respondents indicated that it is necessary to give consent whenever samples are reused for scientific research (66% marked “Definitely yes” or “Rather yes”). Around 42% said strongly that re-consent is necessary when a commercial entity (e.g., a pharmaceutical company) asks the biobank to provide samples. About one-third strongly supported the requirement for re-consent if the subsequent research addresses other scientific problems (35%), or if researchers from other institutions in Poland (36%) or abroad (36%) request that samples are shared. The idea of interpreting the original consent broadly was strongly supported by only 10% of respondents, and moderately by 13% of respondents [38]. Among Polish medical school students, there is a belief that donors should be asked for consent to use their biological material in cases where the samples would be used by external researchers or foreign institutions. Moreover, a lot of students also had opinion that such consent should also be obtained when a new research project is different from the original project. The most of them declared that consent should be acquired before every new research project. Less than one third of respondents stated that no additional consent should be required if the donor consents while donating [37]. In 2023 in Poland, it was shown that 58.8% of respondents believe that they must give their consent to biobank their samples, 34.2% have no opinion on this and 7.0% believe that they do not have to give their consent to biobanking. In addition, 48.0% of respondents believe that they should give their consent every time the samples they provide are to be used for a new research project only 14.2% of respondents consider a one-time consent sufficient. To complete the picture, it is worth noting that in this survey 53.8% of respondents indicated that they can withdraw consent already given and 42.1% did not know if they can make such a decision. Comparing the results with 2010 and 2013, we see that specific consent is much more frequently chosen in Polish society. Respondents are very distrustful of the concept of broad consent. Surprisingly, even medical university students, who are much more aware of biobanking and research using biological material, are reluctant to use broad consent. This may be due to a high sense of autonomy and the need for control over one’s data and donated biological material.

### 4.3. Willingness to Participate in the Study and Factors Affecting the Decision to Donate Biological Material

Willingness to participate in scientific research and willingness to donate biological material was reported in 83% of Finns [43] and 78% of Swedes [44]. It is also worth mentioning that in Finland, two more papers were published in 2019 relating to the willingness to donate one’s biological material to a biobank. In the first of these, respondents, despite having some doubts about donor protection and the protection of personal data, generally have a positive opinion of donating blood to a biobank [45]. In the second, it was shown that Finnish society, despite its openness to scientific research and strong declarations of willingness to donate their biological material to biobanks, has a lower actual willingness to donate biological material than declared [46]. In Italy, 76.3% of students and personnel of the University of Piemonte Orientale that completed the questionnaire agreed to donate their material for research [47]. In Germany 70.4% of the respondents agreed [36]. Among Americans, this percentage was 69% [42]. In contrast, among the Swiss, the percentage was 53.6% [33] and among the Latvians, 36.7% [35]. In 2013 in Poland, less than half of the respondents (45%) were rather ready to donate their sample to the population biobank, and around 12% were definitely ready [38]. In the 2022 survey, approximately half of the respondents declared that they were open to donating their biological material (“rather yes”—30.4%; “definitely yes”—17.2%) [48]. In contrast, in 2023, almost three quarters of Polish medical university students (73.3%) declared a willingness to donate their biological material for research purposes [37]. In our study in Poland in 2023, around 65.3% of respondents agreed to take part in a scientific study that would biobank their biological material and information and their health status. Comparing the results, we see that, in general, more than half of the respondents in Poland would be willing to donate their samples to a biobank. The percentage is higher among medical university students, perhaps due to a greater awareness of biobanking and scientific research in general. Considering the level of knowledge of the respondents regarding biobanking, the percentage of people willing to donate their biological material for biobanking in Poland is exceptionally high. It can be assumed that this is due to a high level of trust in the medical and scientific community.

Citizens of the European Union, on a roughly equal level, were ready to donate information about their genetic profile (34%), medical records (33%), blood samples (30%), samples taken during surgery (30%) or lifestyle information (24%) to the biobank [32]. Latvians in 2019 were willing to donate blood samples (25.5%), samples taken during surgery (26.7%), their genetic profile (27.5%), medical records (35.6%) or lifestyle information (25.8%) [35]. The Swiss would be most willing to hand over their health forms (85.6%), a blood sample (84.6%) and biological samples that they can collect themselves (81.6%). They would be least willing to share information from their social media (14.5%) [33]. Saudi Arabian students would be most likely to donate a blood sample (82%), saliva (77%) or urine (70%) to a biobank [41]. In Poland in 2022, most respondents were willing to donate urine (73.9%), blood (69.7%), hair (69.6%) and tears (69.0%). Respondents were least willing to provide the following biological material: post-mortem donation of brain fragments (20%), sperm (males; 36.4%), egg cells (females; 39.6%) and bone marrow (40.5%) [49]. The results obtained in our study in 2023 do not differ from those obtained by other researchers. In general, in Poland, respondents that are ready to donate their biological material to a biobank are most willing to donate their blood sample (68,4%) or samples they can collect themselves (75%). They are not as willing to share their medical history or lifestyle information. It is also worth noting that respondents generally show a higher willingness to hand over their information than the European average.

Among students in Saudi Arabia, the most important motivating factors for donating their biological material were the opportunity to support the advancement of medical science (44%) and the opportunity to obtain health information (25%) [41]. Among residents of the state of Colorado, the opportunity to contribute to scientific advancement (85%) and support in understanding genetics and disease risk, survival, or treatment methods (72%) were identified as the most important factors influencing decisions to donate a sample to a biobank [42]. Among Swedes, the main motivating factors for donating biological material were the opportunity to support future patients (88.7%) and a personal benefit to the donor or his family (61.1%) [44]. In the cited literature, examples related to altruistic motives predominate, so it is puzzling to see the results presented in this study, where the welfare of future generations is considered by only 36.2% of respondents. In contrast, the opportunity to expand scientific knowledge motivates only 45.4% of respondents. The main motivation of respondents is the opportunity to gain personal benefits and learn something about their own health (62.6%). Comparing these groups is fraught with high risk, as altruistic attitudes may be influenced by a number of factors unrelated to the characteristics of the survey conducted, and factors such as history, education or religion. Although the motivating factors for donating biological material may vary, whether in Poland, Sweden, the United States or Saudi Arabia, the most important factors negatively influencing the willingness to donate biological material to a biobank include concerns about unauthorized use of samples or issues related to confidentiality and sample security [41,42,44]. It is worth noting that in Poland in 2013, the biggest concerns about donating material to a biobank were the use of the material to conduct unethical scientific research and the risk of sharing data with an employer [38]. Despite such high concerns about data security, as many as 15.7% of respondents felt that data should be stored with contact information allowing for immediate identification and 56.2% of respondents indicated that they would prefer that data be stored in a coded manner allowing for possible identification rather than in a fully anonymized manner. This has not changed since 2013; the respondents were more likely to favor the reversible coding of samples than anonymization when asked how to protect samples and donor data [38].

In Poland, in the 2022 study no statistically significant relationships were found between the willingness to donate samples to a biobank and sociodemographic variables, i.e., gender (*t* = 0.734, *p* = 0.486), age (r = 0.052, *p* = 0.087), place of residence (rho = 0.042, *p* = 0.166) or education (rho = 0.001, *p* = 0.994) [48]. In another study from 2022 the potential donor type was identified. The type did not differ statistically in gender, age, education, religiousness or trust in other people [49]. In our study, we also found no statistically significant correlation between the willingness to donate material to the biobank and gender, age, place of residence or education. The most puzzling thing is the lack of the influence of education on the willingness to donate one’s material to biobanking, in all likelihood this is provoked by the total lack of teaching about biobanking whether in elementary school, high school or during college.

### 4.4. Public Trust for Research Using Biological Material and Biobanking

In general, worldwide, scientific institutions are trusted more than governments or commercial or insurance companies. Respondents generally show great concern about the transfer of data from the public sector to the private sector due to low trust in the private sector. They fear that their information could be used to their detriment or in a manner inconsistent with their worldview [39,50,51,52]. In 2010, both in the European Union and in Poland, physicians were given the most trust in terms of access to and oversight of data and samples stored in a biobank, 39% and 44%, respectively [32]. In Latvia, in a 2019 survey the first indication was that physicians were also given the most trust (28.8%) followed by scientists (15.6%) [35]. In Poland, in 2022, a stepwise regression analysis was conducted to define those variables which influenced the most the willingness to donate biological material to a biobank. Willingness to donate biological material to a biobank was best described by trust in scientists (β = 0.082, *t* = 2.134, *p* = 0.033) and doctors (β = 0.080, *t* = 2.089, *p* = 0.037) as well as personal development (β = 0.086, *t* = 2.854, *p* = 0.004) and tradition (β = 0.060, *t* = 1.982, *p* = 0.048) [48]. In the United States, 92% of respondents would be willing to give their data to scientists, 80% to the government and 75% to commercial companies [53]. In Australia, in 2019, 51.8% of the respondents were willing to share their data with the private sector to improve health services and 57.98% for research and development purposes [54]. In Canada, in 2019, respondents had mixed and more negative reactions when there was private sector involvement regarding research based on linked administrative health data [55]. In 2023, a review of the literature found that patients generally expect high transparency and have high concerns about sharing their data with private sector entities [56]. According to our results, Polish society trusts doctors and scientists the most. It is worth mentioning that the most trusted doctors are those whom respondents have direct contact with. The results obtained in our study indicate low levels of trust both in public institutions and commercial companies, which is in some contradiction to the results obtained in other surveys where public institutions tend to have a higher level of trust. This difference can probably be determined by cultural or historical factors. Puzzlingly, compared to other responses from our study, respondents believe that a biobank should be under the supervision of a board of managers rather than a committee of experts (doctors and scientists). This represents some contradiction in terms of information as to whom respondents trust most. However, the source of this contradiction may be an insufficient awareness of how biobanks operate.

## 5. Limitations

Our study has some limitations. The survey was conducted using the CAWI method. Online surveys, due to their characteristics, have their limitations [57,58,59]. It is not possible to reach Internet-excluded people and those who are poorly equipped with new technologies. Another issue is respondent fatigue in the process of answering. During the development of the questionnaire we tried to minimize the number of questions and prepare them in a way that is as understandable and accessible to respondents as possible, however, there is an ongoing risk of respondent fatigue leading to respondents filling out the questionnaire as quickly as possible and answering questions not in accordance with their views. Admittedly, the survey questionnaire was developed based on previously published research results and survey questionnaires, was subjected to expert verification and was tested on a narrow group of people with no ties to scientific research and the biomedical sector, however, these measures may have been insufficient. Unfortunately, due to funding constraints, it was not possible to conduct a pilot study on a smaller study population and consequently create the most refined questionnaire possible. Another issue is the size of the oldest group of respondents. Its size (35.1% of respondents older than 55 years) may possibly influence the results. However, the survey was conducted using a nationwide research panel from Ariadna, which declares it provides anonymized data consistent with a representative distribution of Polish citizens. According to the Central Statistical Office of Poland (CSO), in 2021, citizens aged 18–44 accounted for about 44.8% of all adult citizens. Citizens over the age of 44 accounted for 55.2% of all adult citizens [60]. In our survey, respondents aged 18–44 accounted for 45.2% of all respondents. Older citizens accounted for 54.8% of all respondents. Thus, we believe that the results describe awareness, opinions and attitudes of the Polish population, with a small bias. Finally, we believe that our work may stimulate further research on the topic. Given the above, it seems that conducting an analogous survey, based on an updated questionnaire, using other research methods such as CATI, PAPI or personal interviews could yield even more detailed results. However, it should be borne in mind that such a survey will be much more time-consuming and will generate higher costs.

## 6. Conclusions

In Poland, despite a relatively low level of knowledge among the public about biobanking, respondents showed a surprisingly high willingness to donate their biological material to a biobank. It can be assumed that the Polish public has a high level of trust in medical and scientific personnel working at universities and public research institutions for this reason; despite the low level of awareness, they are ready to participate in scientific research. It is also worth noting that the results indicate that the main motivation for participating in scientific research is personal or family benefits. The biggest concerns that respondents have are related to issues of access to the data they provide and the safeguards used. Researchers wishing to recruit potential donors during their meeting should focus on indicating what the potential personal benefits are that such a donor may gain from donating their biological material, and should explain in detail how they intend to ensure the safety of the samples and data collected. There is also an apparent need for educational activities aimed at informing the public about scientific research with a special focus on biobanking. It is also crucial to raise awareness about the role that informed consent plays in the entire process and what rights a potential donor has. Given the low trust in the government and the pharmaceutical industry, it makes sense that the awareness-raising process should be led by the medical or scientific community. It would seem appropriate to run an information campaign, featuring representatives of the scientific or medical community who are well known and trusted by society, to raise the public awareness of biobanking and inform the public about the advantages and benefits of the process.

## Figures and Tables

**Figure 1 healthcare-11-02714-f001:**
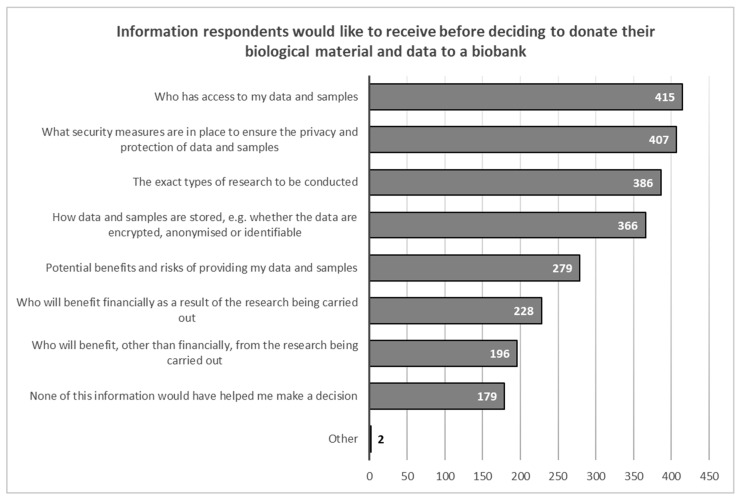
Overview of information respondents would like to receive before deciding to share their biological material and data with a biobank.

**Figure 2 healthcare-11-02714-f002:**
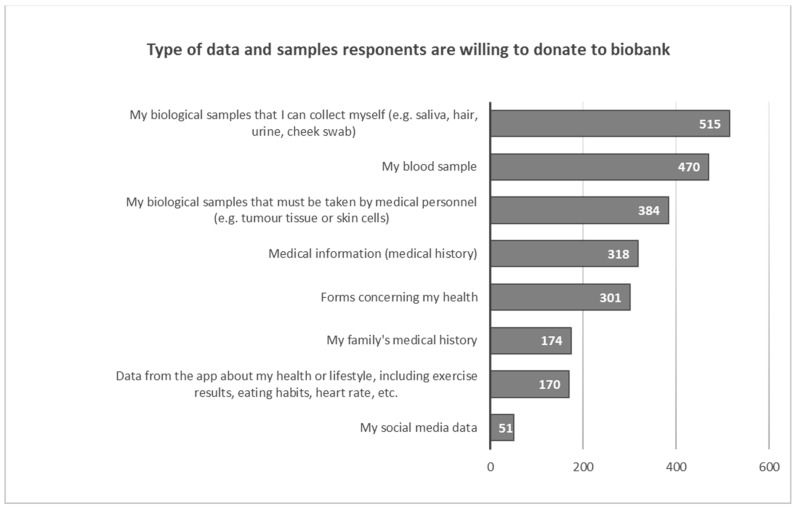
Willingness to donate specific biological material or data.

**Figure 3 healthcare-11-02714-f003:**
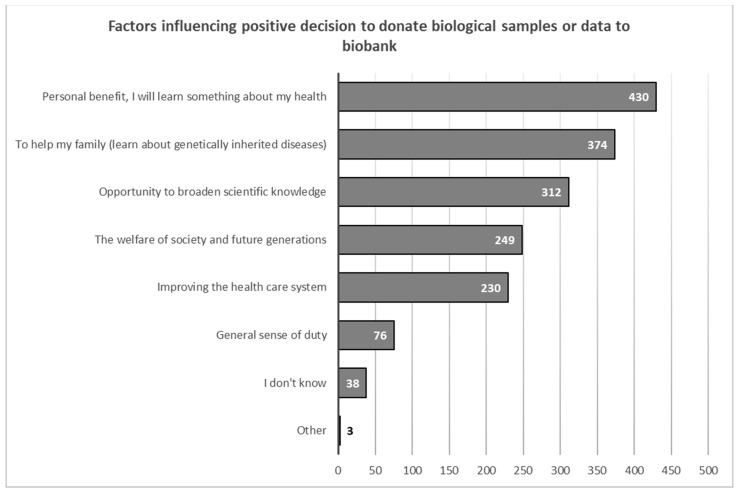
Factors influencing a positive decision regarding biobanked information in a study.

**Figure 4 healthcare-11-02714-f004:**
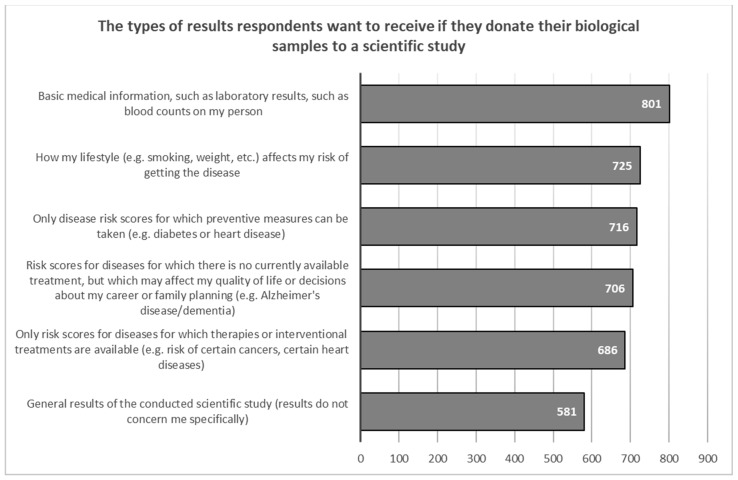
Willingness to receive distinct types of research results.

**Figure 5 healthcare-11-02714-f005:**
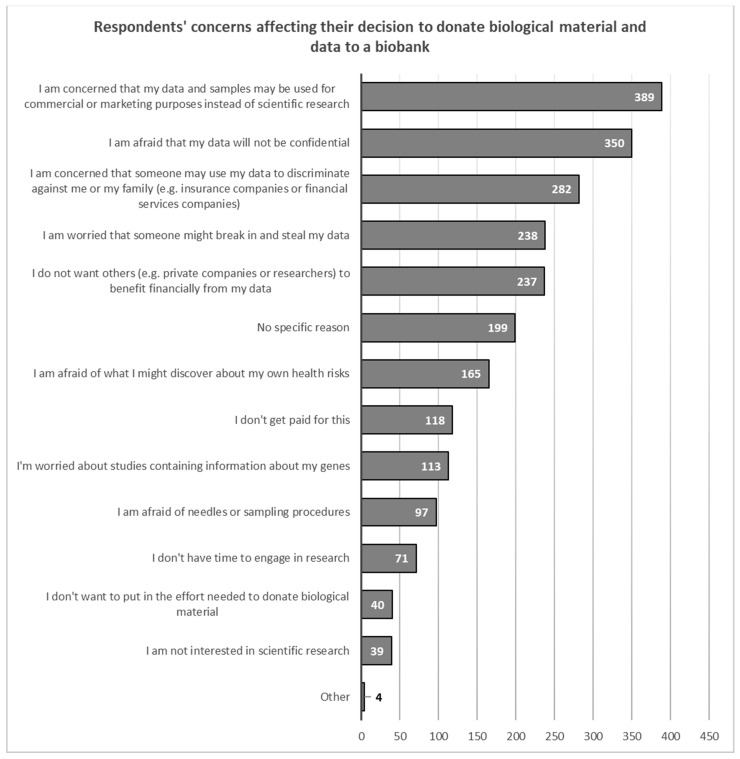
Overview of concerns that respondents have regarding their decision to share the biological samples and data with a biobank.

**Table 1 healthcare-11-02714-t001:** Socio-demographic characteristics of respondents.

Variable	Category	Result
Number of respondents *N*		1052
Sex *N* (%)	Female	567 (53.9)
	Male	485 (46.1)
Age *N* (%)	18–24	106 (10.1)
	25–34	205 (19.5)
	35–44	170 (16.2)
	45–54	202 (19.2)
	55 years or more	369 (35.1)
Age mean (SD)		46.37 (15.92)
Place of residence *N* (%)	Rural area	390 (37.1)
	Up to 20,000 residents	138 (13.1)
	20,001–50,000 residents	94 (8.9)
	50,001–100,000 residents	111 (10.6)
	100,001–200,000 residents	84 (8.0)
	200,001–500,000 residents	101 (9.6)
	Above 500,000 residents	134 (12.7)
Education *N* (%)	Primary or Junior high school	127 (12.1)
	High school	366 (34.8)
	Post-high school	115 (10.9)
	Bachelor’s degree	87 (8.3)
	Master’s degree	357 (33.9)
Marital Status *N* (%)	Singel	310 (29.5)
	Married or partnership	573 (54.5)
	Divorced	108 (10.3)
	Widowed	61 (5.8)

**Table 2 healthcare-11-02714-t002:** Awareness of biobanking.

Variable	Yes *N* (%)	No *N* (%)	Don’t Know *N* (%)
I have heard of scientific research using biological samples	701 (66.6)	205 (19.5)	146 (13.9)
I have heard of biobanking before	220 (20.9)	674 (64.1)	158 (15.0)

**Table 3 healthcare-11-02714-t003:** Respondents’ opinion on scientific research using biological samples.

Opinion on Scientific Research Using Biological Samples (Such as Blood, Saliva, Urine, Hair or Tissues)	Results *N* (%)
Negative	18 (1.7)
Rather negative	50 (4.8)
Rather positive	441 (41.9)
Positive	172 (16.3)
Difficult to say/I have no opinion	371 (35.3)

**Table 4 healthcare-11-02714-t004:** Definition of biobanking according to respondents.

Definition	Results *N* (%)
Biobanking is the process by which body fluid or tissue samples, genetic data and medical data (medical history, laboratory results, etc.) are collected and stored in order to better understand health and disease progression.	441 (41.9)
Biobanking is the process by which seed samples are collected to safeguard them in the event of a global environmental disaster.	133 (12.6)
Biobanking is a process in which data on a person’s social and financial situation and medical data (medical history, laboratory results, etc.) are collected and stored in order to better understand the impact of financial situation on health status.	56 (5.3)
None of these	32 (3.0)
I don’t know	390 (37.1)

**Table 5 healthcare-11-02714-t005:** Awareness of respondents regarding consent in biobanking.

Variable	Yes *N* (%)	No *N* (%)	Don’t Know *N* (%)
Donor consent is needed for biobanking	619 (58.8)	73 (6.9)	360 (34.2)
Donor may withdraw consent	566 (53.8)	43 (4.1)	443 (42.1)

**Table 6 healthcare-11-02714-t006:** Type of consent preferred by respondents.

Type of Consent	Specific Consent	Tiered Consent	Broad Consent	No Opinion
Result *N* (%)	505 (48.0)	188 (17.9)	149 (14.2)	210 (20.0)

**Table 7 healthcare-11-02714-t007:** Respondents willingness to donate biological samples.

Variable	Yes *N* (%)	No *N* (%)
Willingness to donate biological samples	687 (65.3)	365 (34.7)

**Table 8 healthcare-11-02714-t008:** Ownership of samples and data.

The Owner of the Data and Samples Found in the Biobank Should Be:	Results *N* (%)
Me personally	562 (53.4)
Biobank	310 (29.5)
No opinion	165 (15.7)
Specific scientists who conduct research and make discoveries	161 (15.3)
Universities or organizations co-founding the biobank	84 (8.0)
Government	17 (1.6)
Other	3 (0.3)
No one	29 (2.8)

**Table 9 healthcare-11-02714-t009:** Level of trust.

Variable	1—Lack of Trust *N* (%)	2 *N* (%)	3 *N* (%)	4 *N* (%)	5—Full Trust *N* (%)
My doctor/General Practitioner	56 (5.3)	73 (6.9)	320 (30.4)	390 (37.1)	213 (20.2)
Doctors in general	106 (10.1)	143 (13.6)	432 (41.1)	307 (29.2)	64 (6.1)
Researchers at the university	81 (7.7)	119 (11.3)	396 (37.6)	357 (33.9)	99 (9.4)
Researchers in other public institutions	91 (8.7)	155 (14.7)	409 (38.9)	312 (29.7)	85 (8.1)
Pharmaceutical companies	298 (28.3)	264 (25.1)	353 (33.6)	116 (11.0)	26 (2.5)
Other than pharmaceutical, global, for-profit private companies	464 (44.1)	257 (24.4)	251 (23.9)	70 (6.7)	10 (1.0)
Other than pharmaceuticals, Polish for-profit private companies	477 (45.3)	253 (24.0)	256 (24.3)	53 (5.0)	13 (1.2)
Insurance companies	461 (43.8)	255 (24.2)	254 (24.1)	67 (6.4)	15 (1.4)
Government	581 (55.2)	185 (17.6)	194 (18.4)	73 (6.9)	19 (1.8)

**Table 10 healthcare-11-02714-t010:** Responsibility for storage and management of samples and data.

Responsibility for Samples and Data Stored in the Biobank Should Rest with the:	Results *N* (%)
Biobank (Management Board)	567 (53.9)
Independent expert committee (e.g., independent researchers: scientists and clinicians not associated with the biobank)	337 (32.0)
An independent committee representing the public (e.g., citizens, patients, the public)	155 (14.7)
Mixed Committee composed of the public and experts	126 (12.0)
Other	17 (1.6)

**Table 11 healthcare-11-02714-t011:** Correlation between socio-demographic factors and awareness of biobanking or willingness to donate biological material.

Variable	Awareness of Scientific Research ρ	Knowledge of Term Biobanking ρ	Opinion on Scientific Research ρ	Willingness to Donate Biological Material ρ
Age	−0.002	0.035	−0.124	−0.042
Education	−0.112	−0.119	−0.093	−0.068
Place of residence	−0.045	0.006	−0.057	−0.051
Gender	−0.023	0.027	−0.021	−0.021

The table shows the correlation coefficients calculated via Spearman’s Q method.

**Table 12 healthcare-11-02714-t012:** Correlation between awareness, knowledge and opinion of biobanking and willingness to donate biological material.

Variable	Willingness to Donate Biological Material ρ	*p*-Value
Opinion on scientific research	0.475	<0.05
Awareness of scientific research	0.211	<0.05
Knowledge of term biobanking	0.136	<0.05

The table shows the correlation coefficients calculated via Spearman’s Q method and the *p*-value.

## Data Availability

The datasets used and/or analyzed during the current study are available from the corresponding author upon reasonable request.

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
