# Peer review of "Awareness, Attitudes and Willingness to Donate Biological Samples to a Biobank: A Survey of a Representative Sample of Polish Citizens"

_healthcare, 2023, doi:10.3390/healthcare11202714_

Round 1

Reviewer 1 Report

Dear Authors,

Thank you very much for the opportunity to review your paper.

The article itself and the topics discussed in it deserve attention and I believe that constitute a very important voice in the discussion on biobanking of biological samples. I congratulate the Authors on an interesting study, but before publication, I would like to draw attention to certain details, that should be corrected before the article is published.

1. I encourage you to read the article again from a linguistic perspective, because in some places it contains certain formulations that seem incorrect to me. E.g. lines: 91-92 or 215-216. Moreover, Table 1 contains Polish words that should be replaced with English equivalents ('mieszkańców'). You also don't start a sentence with a number, which happens often in this text.

2. I encourage you to review the work again from an editorial perspective. For example, Table 1 in the description part there is: Age (mean (SD)), while the values are given differently, because 46.37 (15.92), so the average is without brackets. I think it is worth reformattin. 

Table 4. Perhaps the table would look better if the questions were arranged vertically (they are quite long, so graphically the table does not look good). Figure 1 - has two titles (different), one in the object area and the other in the caption. It is worth unifying it and keeping one, more appropriate according to the authors. I also don't understand why the answers in Figures are written in block letters, it's tiring for the reader.

Despite these minor comments, I evaluate the work positively and my suggestions are supportive. However, I ask the authors to consider them.

They are all mentioned in the section: Comments and Suggestions for Authors

Author Response

Dear Reviewer, 
thank you very much for your work and a number of valuable comments. Referring to your comments:
1. The article has been revised in terms of language. We have made changes to make the content clear and unambiguous. We have corrected linguistic errors and mistakes that crept in during translation. 
2. In accordance with your comment, we have analyzed the document from an editorial point of view. We standardized the presentation of data. We modified Table 4 according to your recommendation, made the naming in Fig. 1 more consistent, and changed of block letters to lowercase.

In addition, based on the advice of the other Reviewers, we expanded the introduction to the paper and diverged the discussion. 

Again, thank you very much for your comments 

Reviewer 2 Report

The research tackles important topic of the awareness, attitudes and willingness to donate biospecimens for biobank research among Polish citizens. I find the paper interesting, important and timely, especially that to date there are only few works on the topic conducted in Polish population (unfortunately, the Authors have failed to address any of those research, which I discuss below). Another advantage of this study is that it was conducted on a large and representative sample. This in turn, provided new in-depth knowledge about the attitudes of Polish society on biobank donation. However, while I believe that this research can be of interest to the readers of the Journal there are some major issues that have to be revised before it could be considered for published. Below I list the main points:

Abstract:

According to the Journal’s guidelines the abstract should be a single paragraph and should follow the style of structured abstracts, but without headings.

Additionally, the abstract should provide more detailed information on main results.

Key words

Add the following key words: awareness, attitudes, tissue donation,

Introduction:

 To make this section more readable divide it into several paragraphs.

Some of the Authors’ comments require justification or reference. For example, in the very first sentence the Authors declare: “With the COVID - 19 pandemic, attitudes toward obtaining and sharing medical data 30 have changed significantly in Europe and around the world”. How do you know that. Can you show any data that confirm such conclusion?

What is the justification for the first few sentences about COVID-19 pandemics? (lines 30-38). I do not see the point to mention COVID-19 at all. Instead, I would recommend focusing on biobanking.

Describe the main international organizations in the field of biobanking (BBMRI-ERIC, ISBER, their commitment to biobanking and the main international science outreach initiatives that are already being carried out (e.g., European Biotech week, part of the international Global Biotech week initiative, and others), the importance of donors’ involvement. Next, provide more information on the description of biobanking in Poland.

While describing biobanking in Poland the Authors should cite relevant literature, which is missing:

-- Witoń M, Strapagiel D, Gleńska-Olender J, Chróścicka A, Ferdyn K, Skokowski J, Kalinowski L, Pawlikowski J, Marciniak B, Pasterk M, Matera-Witkiewicz A, Kozera Ł. Organization of BBMRI.pl: The Polish Biobanking Network. Biopreserv Biobank. 2017;15(3):264–69.

-- Kozera Ł, Strapagiel D, Gleńska-Olender J, Chrościcka A, Ferdyn K, Skokowski J, Kalinowski L, Pawlikowski J, Marciniak B, Pasterk M, Matera-Witkiewicz A, Lewandowska-Szumieł M, Piast M, Witoń M. Biobankowanie ludzkiego materiału biologicznego dla celów naukowych w Polsce i w Europie. Stud Iurid. 2018;73:13–28.

-- Chróścicka A, Paluch A, Kozera Ł, Lewandowska-Szumieł M. The landscape of biobanks in Poland - characteristics of Polish biobanking units at the beginning of BBMRI.pl organization. J Tran Med. 2021;19(1):267. https://doi.org/10.1186/s12967-021-02926-y.

-- Krekora-Zając D. Legal aspects of biobanking HBS for scientific purposes in Poland. Studia Prawnicze. 2019;4(220):165–84.

Lines 53-57: some more information on ethical, legal, and social issues (ELSI) related to the acquisition, storage, and sharing of biosamples donated for biobank research, especially in terms of data protection, privacy and confidentiality, as well as the commercialization of research results should be discussed. Additionally, the Authors should cite relevant literature. See:

-- Hoeyer KL. Size matters: The ethical, legal and social issues surrounding large-scale genetic biobank initiatives. Nor. Epidemiol. 2012, 21(2), 211–220. https://doi.org/10.5324/nje.v21i2.1496.

-- Bledsoe MJ. Ethical, legal and social issues of biobanking: past, present, and future. Biopreserv Biobank. 2017, 15(2), 142–147. https://doi.org/10.1089/bio.2017.0030.

-- Caulfield T. and  Murdoch B. Genes, cells, and biobanks: Yes, there's still a consent problem. PLoS Biol. 2017, 15(7), e2002654. https://doi.org/10.1371/journal.pbio.2002654.

-- Tzortzatou-Nanopoulou O., et a. Ethical, legal, and social implications in research biobanking: A checklist for navigating complexity. Developing World Bioethics, 2023. https://doi.org/10.1111/dewb.12411

Methods

What, in my opinion, would be worth working on is the structure of the methodological part. For better reception, it is worth dividing the content of the methodological part according to the following scheme: study design, participants and setting, research tools, data collection, ethical issues, data analysis. In its present form, some of these aspects are mixed or omitted, which reduces the methodological value of the work.

What is: Ariadna company? A professional survey company?

Was the Polish tool validated? If not it should be mentioned in limitations.

Results:

Table 1 has to be revised:

-- Some Polish words are present “Mieszkańców”

-- “Large city” appears three times. It is confusing. I suggest replacing it with the following scale:

Place of residence

Up to 10,000 residents

10–50,000 residents

51–100,000 residents

51–100,000 residents

101–500,000 residents

Above 500,000 residents

-- Marital Status: everything is mixed here. Put these categories in the following order:

Single

A partnership

Married

Widowed

Divorced

-- Education: Why there is no education higher than bachelor? No University?

Additionally, taking under consideration educational system in Poland what is: Post-secondary school”? Again, cafeteria of responses should be put in order from the lowest to the highest level.

It would be interesting to compare responses from those who are biobank aware (have heard about biobanks before) and those who are not (have not heard about biobanks before)

Table 3: The term “Narrow Consent” rarely appears in the literature? Do you mean “specific consent” (for one experiment with well-defined aim / before every research that involves my samples), “tiered consent (individually selected categories of research or research uses e.g. specific diseases, i.e. cancer or neurological diseases, or research conducted only by specified institutions, i.e. publicly-funded but not private) or other type of consent?

Did you use those names in your research? Research show that the general public possess limited (if any) knowledge about research ethics and may not be familiar with the models of consent used in biomedical research. Thus, using such terms in your questionnaire could be misleading.

Respondents’ opinions on the biobank definition should be discussed after Table 2.

Fig. 1. First present declarations on the willingness to donate biospecimens and that related data

Fig. 2. and 4: Change the order of answers: Other should be always placed at bottom.

Fig. 5 should be placed earlier before or after preferred type of consent.

Table 5. Again, order responses: Put “No one” after all other specific subjects, “No opinion” as the last possible answer

Most importantly, the Authors should provide at least basic analysis of correlations between sociodemographic variables from Table 1 and various topics. Since research like this are important for biobanks and researcher as they are often wondering who can be recruited as a possible donor. Without such analysis the readers can ask themselves: “so what?”.

Discussion

The literature in the Discussion section should be updated as there are many new research on the attitudes towards human tissues donation for research purposes.

More importantly, it is surprising that the Authors do refer to any of previously published research conducted in Poland:

-- Pawlikowski, J. Biobankowanie ludzkiego materiału biologicznego dla celów badań naukowych – aspekty organizacyjne, etyczne, prawne i społeczne. Wydawnictwo Uniwersytetu Medycznego w Lublinie, Lublin 2013.

-- Pawlikowski, J.; Wiechetek, M.; Majchrowska, A. Associations between the willingness to donate samples to biobanks and selected psychological variables. Int. J. Env. Res. Public Health. 2022, 19(5), 2552. https://doi.org/10.3390/ijerph19052552.

-- Majchrowska, A.; Wiechetek, M.; Domaradzki, J.; Pawlikowski, J. Social differentiation of the perception and human tissues donation for research purposes. Front. Genet. 2022, 13, 989252. https://doi.org/10.3389/fgene.2022.989252.

--Domaradzki, J.; Czekajewska, J.; Walkowiak, D. To donate or not to donate? Future healthcare professionals’ opinions on biobanking of human biological material for research purposes. BMC Med Ethics. 2023, 24(1), 53. https://doi.org/10.1186/s12910-023-00930-z.

Thus, the Authors should compare their interesting findings and show how do they differ from previous Polish studies on the topic.

 This section should end with “Limitations”.

Conclusion

The Conclusion section should be more reflective of policy implications of the findings. For example, the Authors could reflect on what systemic approaches should be undertaken to promote the awareness and interest in biomedical research and donation for research purposes.

To conclude, while in my opinion the issues raised in this manuscript are important and timely, and the paper itself fits well with the aims of Journal, I also believe that it requires some major revisions and completion before it could be published.

Author Response

Dear Reviewer,

Thank you very much for your work and your series of comments on the content of the manuscript. Your comments are valuable to me and allow me to refine the content of the publication. I have addressed each of your comments below and provided an explanation for them and indicated what changes I made to the content of the manuscript. 

Abstract:

In preparing the abstract, I based my work on others abstracts available in already published papers in the Healthcare journal. For this reason, sections such as "Background"; "Methods" or "Conclusion" remained in the content. I believe that your comment is correct and I have modified the content of the abstract so that it is in accordance with your recommendation. I have also expanded the results section. I hope you will find it sufficient. 

Key words

Thank you very much for your comment, I have added the relevant key words, in the manuscript.

Introduction:

I rewrote the introduction to the manuscript according to your comments and recommendations. I divided it into 3 sections: the first on international organizations involved in biobanking, the second on biobanking in Poland, and the third on ethical, legal and social aspects of biobanking. I used all the literature you indicated. . I also resigned from referring to the COVID-19 pandemic.

Methods

I modified the methodological part of the manuscript and included the sections you indicated. I hope that now the document presents the information in a standardized and accessible way for the reader.

Ariadna is nationwide independent survey research panel company in Poland, that specializes among others in Computer Assisted Web Interviews (CAWI). Ariadna carry out nationwide studies, surveys, and experimental research fully embracing the highest standards of rigour and integrity required of scientific work.

In survey only genuine persons of a verified identity can take part. Ariadna decisively excludes carrying out research using random sampling methods, where respondents are recruited using pop-ups and banners on web pages displayed to unknown persons or using mass mailing to unverified email address databases and various kinds of internet polls in social media, for instance. Ariadna ensures that only sound and reliable research is carried out.

All research conducted by Ariadna panel respects and observes the ethical principles and legal regulations concerning the conduct of market research and opinion polling as well as the GDPR legislation on personal data protection. I find your comment very important so I’ve added more clarification to the manuscript. I believe on you and potential readers to find it sufficient.

The Ariadna panel has a current and valid Interviewer Quality Control Program (PKJPA) certificate confirming the high quality of the research services provided, which is issued on the basis of an independent audit carried out annually by the Polish Association of Public Opinion and Marketing Research Firms (www.pkjpa.pl). Other companies with PKJPA certification include are for example Centre for Public Opinion Research (CBOS), AC Nielsen Poland, Kantar Poland or IPSOS Poland.

Results:

I modified Table 1 according to your recommendations. I sorted the data in ascending order and set appropriate ranges. In terms of education, I modified the nomenclature so as to properly reflect the specificity of the Polish school system. I have taken into account your recommendations for martial status. Our data did not allow us to differentiate between marriages and partnerships and for this reason we included them in one group. Unfortunately, due to time constraints, we were not able to compare the responses of people who had previously heard of biobanks against those who had not heard of them.

By using the term narrow consent, I meant using the term specific consent. I have changed the nomenclature in the body of the manuscript. During the study, we did not ask participants for their opinion using specialized expressions. We tried to describe each type of consent in simple language that was as easy as possible for people not involved in biobanking or biomedical research. We also verified the content of the questionnaire among people not involved in scientific activities. They claimed to understand the questions presented.

I have modified the tables, figures and data layout according to your recommendations. I have also added correlation analysis in the body of the manuscript.

Discussion

I have expanded the discussion section in line with your recommendations. I have included all the publications you indicated. We have compared in more detail the results obtained in our study to those of other studies conducted in Poland. I’ve added section “Limitations”

Conclusion

We have developed the conclusion section to give a better indication of the kind of action that should be taken to raise awareness of biobanking and scientific research in Polish society.

Reviewer 3 Report

Dear authors,

You have decided to gauge Polish citizens' awareness and attitudes towards biobanking and their willingness to donate biological samples and presented answers from the questionnaire. It is popular and very user friendly article for the huge specter of people but without big scientific value as you know. The results are expected.

First of all, the article is not well written, you have only 3 references in the introduction and 17 references in total. This means that the literature cited is below any reasonable level. You have cited similar articles and you can see what a well cited article looks like. This is not the case with your article, you have to rewrite and recite the entire article.

Second, you mentioned a correlation analysis that was done. However, this is not presented anywhere in the entire text. You have provided us with a simple descriptive statistic with percentages IN which in my opinion is not sufficient for publication.

Where are the limitations of a study, this type of article deserved a paragraph about it.

Notably, most of the participants were over 55 years old, which greatly affected the results. The population is  not evenly distributed in terms of age.

Minor editing of English language required.

Author Response

Dear Reviewer, 
Thank you very much for your comments on our manuscript. They are very valuable to us, reading them will enable us to refine our paper even better. 
In accordance with your remark, we approached the writing of the article again, developed the introduction and divided it into 3 sections. It is currently supported by more than 30 literature references. We have analyzed the methods and results sections and reorganized the presentation of information so that they form a more logical and coherent whole. We have expanded the discussion by referring to more papers describing awareness and willingness to donate samples in Poland. We also presented the results of the correlation analysis and described the limitations of our study.

Referring to the issue of the age of the respondents. Thank you very much for your comment, however, we cannot agree with it. The survey was conducted using a nationwide research panel Ariadna, which declares to provide anonymized data consistent with a representative distribution of polish citizens. According to the Central Statistical Office, in 2021, citizens aged 18-44 accounted for about 44.8% of all adault citizens. Citizens over the age of 44 accounted for 55.2% of all adult citizens. In our survey, respondents aged 18-44 accounted for 45.2% of all respondents. Older citizens accounted for 54.8% of all respondents. In our opinion, the distribution obtained in the survey is in line with the current distribution in Poland. 

Round 2

Reviewer 2 Report

I have read both Authors’ response and the revised manuscript itself with interest. The Authors have clarified most issues raised in the review and I believe that this revised manuscript is now more consistent owing to their corrections and additional arguments. On the whole, I appreciate this effort. Ath the same time, still there are some additional minor issues that should be improved:

1. While the Authors have improved the number of references the list of the research on the public attitudes on biobanks should be updated, as they cite some older studies, i.e. Goodson and Vernon 2004; Kaufman et al. 2009; Gottweis et al 2011.

2. References are prepared careless and do not follow the Journal’s guidelines. Additionally No 18 and 19 basically the same. Why put it twice?

3. Still, I wonder why although the study participants were divided into age groups by decades, the last group (aged above 55) which is the most numerous (35.1%) was not divided accordingly? This is confusing and possibely infuelces study results.

4. The Conclusion section could be shortened. Additionally  I do not believe that some of recommendations provided by the Authors are either justified or possible to implement. For example: “changing the primary or high school curriculum”. Since issues related to biomedical research, biobanking and ELSI are often too complex for young people I woudl suggest launching information biobank campaigns that would rise public’s awareness on biomaterial donations instead.

 All in all, I believe that the research has been much improved and with some additional minor revisions can be accepted for publication.

Author Response

Dear Reviewer,

thank you very much for your comments. They significantly help me in manuscript preparation. Below I have provided responses to the points you raised in the review.

Ad 1.

I have updated the references in the manuscript to refer to more current research. I have decided to resign from quoting Goodson and Vernon 2004 paper, instead I’ve compared our results with study conducted in Italy by Aleni C et all. in 2022. Comparing the results with a European country is, in my opinion, much more reliable due to smaller differences in society and the fact that both countries are bound by the European legal order. I’ve allowed myself to leave the Kaufman et. al. paper from 2009 due to the large research group, but added two new papers relating to level of trust regarding health data sharing with private (Braunack-Mayer et.al. 2021 and Paprica et.al. 2019). I also resign from quoting Gottweis et. al. 2011 paper, instead referred to papers describing public trust in health system and concerns regarding public-private cooperation (Papoutsi et.al.2015 and Kerasidou et.al. 2023). In addition, I have added two more papers relating to the willingness to donate biological material in the Finnish society to refer to more actual knowledge (Raivola et. al. 2019 and Snell et. al. 2019).

Ad 2.

Thank you very much for your comment. I have corrected the references so that they comply with the requirements of the journal. I have changed the descriptions of references 18 and 19 so as to differentiate them more. Reference 18 refers to the mission and specificity of PNB, while reference 19 is statistical information on the total number of biobanks in PNB.

Ad 3.

Unfortunately, based on the data we have, we are not yet able to separate the oldest group of the respondents in to the 3 more groups 55-64, 65-74, 75+. The survey was conducted using a nationwide research panel Ariadna, which declares to provide anonymized data consistent with a representative distribution of polish citizens. According to the Central Statistical Office, in 2021, citizens aged 18-44 accounted for about 44.8% of all adault citizens. Citizens over the age of 44 accounted for 55.2% of all adult citizens. In our survey, respondents aged 18-44 accounted for 45.2% of all respondents. Older citizens accounted for 54.8% of all respondents. In my opinion, the distribution obtained in the survey is in line with the current distribution in Poland, however, I understand your comment that this may have affected the results and have therefore added an additional paragraph in the limitations section to clearly describe the whole situation and to inform potential readers about this issue.

Ad 4.

As per your recommendation, I have shortened the conclusion section. I have removed the proposal to make changes to the curriculum. I agree with your opinion that this knowledge may be too complicated for elementary or high school students. I also removed the reference to legal changes, I continue to believe that they are necessary, nevertheless the manuscript refers to public awareness and attitudes regarding biobanking and not the legal status. As you suggested, I pointed out the need for an information campaign.

I hope I was able to clarify any issues of concern you raised during the review.

Reviewer 3 Report

Dear authors, thank you for deciding to redesign your manuscript. In general, I think you are now more satisfied with your new version as well. Your comment regarding the age is well described. I would only add that you should add an explanation under or around Table 11 and 12. In Table 1 all data are correlation coefficients and in Table 12 we have coefficients and p-values.
Regards

Author Response

Dear Reviewer,

thank you very much for your comments. In accordance with your recommendation, I have added the appropriate explanation to Table 11 and Table 12. I would also like to inform you that, following the comments of another Reviewer, I have updated the references to be more up-to-date and shortened and modified the manuscript conclusion section.